# *Kalmusia variispora* (Didymosphaeriaceae, Dothideomycetes) Associated with the Grapevine Trunk Disease Complex in Cyprus

**DOI:** 10.3390/pathogens14050428

**Published:** 2025-04-28

**Authors:** Georgios Makris, Marcelo Sandoval-Denis, Pedro W. Crous, Loukas I. Kanetis

**Affiliations:** 1Department of Agricultural Sciences, Biotechnology and Food Science, Cyprus University of Technology, 3036 Limassol, Cyprus; gp.makris@edu.cut.ac.cy; 2Westerdijk Fungal Biodiversity Institute, Uppsalalaan 8, 3584 CT Utrecht, The Netherlands; m.sandoval@wi.knaw.nl (M.S.-D.); p.crous@wi.knaw.nl (P.W.C.)

**Keywords:** Cyprus, Didymosphaeriaceae, grapevine trunk diseases, *Kalmusia variispora*, *Vitis vinifera*

## Abstract

Grapevine trunk diseases (GTDs) are widespread worldwide, causing serious economic losses to the vitiviniculture industry. The etiology of the complex pathogenic mycobiome associated with this group of diseases is critical to implementing appropriate management strategies. Diseased grapevines exhibiting typical GTD symptoms were collected from vineyards in different provinces of Cyprus, resulting in 19 pycnidial isolates. A subsequent multilocus sequence analysis of six genetic loci (ITS, LSU, SSU, *b-tub*, *tef1-a*, and *rpb2*) identified them as *Kalmusia variispora*, and twelve representative isolates are included in the phylogenetic analyses. According to pathogenicity trials on two-year-old potted vines (cv. Mavro), all tested isolates were pathogenic, exhibiting light to dark brown discoloration and lesions of varying levels, ranging from 4 to 12.3 cm long. The capacity of *K. variispora* isolates to produce cell-wall-degrading exoenzymes was qualitatively estimated on solid media. Cellulase, pectinase, and laccase production were evident for all the tested isolates, except isolate CBS 151329, where the latter enzyme was undetected. The severity of the symptoms was consistent with the laccase-producing capacity. The present study confirmed the association of *K. variispora* with grapevines as a pathogen and represents the first description of this ascomycete as a GTD causal agent in Cyprus. This highly virulent species may play a significant role in the GTD complex, and its biological cycle and epidemiology should be further investigated.

## 1. Introduction

Grapevine trunk disease (GTD) is a collective term used to house a group of fungal diseases responsible for an array of symptoms associated with vascular infections [1,2,3,4]. Although GTDs are inextricably associated with *Vitis vinifera* [5,6], during the last three decades they have re-emerged, with a dramatic impact on the longevity and the profitable vineyard lifespan, posing a considerable threat to the sustainability of the vitivinicultural sector worldwide [7,8,9].

Numerous wood-inhabiting fungal species have been associated with GTDs, initiating infections of the perennial plant organs via wounds, and mostly induced during the annual pruning process or other cultural and propagation practices [10,11]. The predominant GTD-associated, pathogenic taxa comprise the ascomycetous species *Eutypa lata* and *Phaeomoniella chlamydospora*, numerous species in the Botryosphaeriaceae, Diaporthales genera of *Phaeoacremonium*, *Ilyonectria*, *Dactylonectria*, and *Cadophora*, and species in the basidiomycetous taxa of *Fomitiporia*, *Phellinus*, and *Stereum* [5,6,12]. However, recent research advances, as well as grapevine microbiome studies, have increased our knowledge of the fungal species related to the GTD complex [13,14,15,16,17]. Beyond well-known GTD pathogens, other fungal taxa have been found in association with GTDs, such as *Biscogniauxia rosacearum* [18], *Macrophomina phaseolina* [19,20], *Truncatella angustata* [11,21], *Neofabraea kienholzii* [22], and *Neosetophoma italica* [23], and numerous species of *Cytospora* [24,25], *Seimatosporium* [10,26,27], *Kalmusia* [28,29], *Paraconiothyrium* [16,30], *Pestalotiopsis*, and *Neopestalotiopsis* [4,31,32,33], as well as *Fusarium* and *Neocosmospora* spp., which are associated with young vine decline [34].

The genus *Kalmusia* is placed in Didymosphaeriaceae (=Montagnulaceae) [35] and typified by *K. ebuli* Niessl., which was recently neotypified due to the loss of the original specimen [36]. Verkley et al. established the asexual genus *Dendrothyrium*, with *D. variisporum* (CBS 121517^T^) as the species type, and described *D. longisporum* (CBS 582.83^T^) [37]. However, Ariyawansa et al. later reported that these two species clustered together with *K. ebuli*, leading to the synonymy of *Dendrothyrium* under *Kalmusia* [35]. According to Zhang et al., more than 40 species belong to *Kalmusia* [36]. However, living ex-type strains and DNA sequence data are available for only eight species (i.e., *K. araucariae*, *K. cordylines*, *K. ebuli*, *K. erioi*, *K. italica*, *K. longispora*, *K. spartii*, and *K. variispora*). Species within *Kalmusia* are considered mainly saprobic, isolated from a variety of terrestrial habitats, such as leaves, stumps, and dead branches of shrubs and trees [36,37,38,39,40,41,42,43], also exhibiting endophytic features [13,44]. To the best of our knowledge, only two species of the genus, *K. longispora* and *K. variispora*, have been reported as causal agents of plant diseases, mostly associated with diebacks of perennial organs of woody host plants [23,28,29,45,46,47].

Plant pathogenic fungi employ a diverse secretome to depolymerize the main structural polysaccharide components of the plant cell wall, i.e., cellulose, hemicellulose, and pectin [48]. However, there is evidence of significant differences in the importance of specific cell-wall-degrading enzymes or families of enzymes even between species from the same genus [3,49]. Furthermore, recent advances in the research of major trunk disease-associated fungi suggest that the destructive colonization of grapevine tissues employ a variable arsenal of cell-wall-degrading enzymes and phytotoxic secondary metabolites that contribute to host damage and symptom development [49,50].

During a field survey in the local vineyards in Cyprus, numerous *K. variispora* isolates were collected, in association with GTD-related symptoms. Our objectives are (a) to provide a morphological description and phylogenetic analyses of this under-studied GTD-associated species in Cyprus and (b) to evaluate its potential as pathogen under local conditions.

## 2. Materials and Methods

### 2.1. Sampling and Fungal Isolation

Isolates used in this study were collected from 15 vineyards (8 to 70 years old) of wine-grape and table-grape cultivars (Cabernet Sauvignon, Mavro, Shiraz, Superior, and Xynisteri) expressing typical GTD symptoms between 2017 and 2018 in the provinces of Nicosia, Limassol, and Paphos, Cyprus (Appendix A). Wood samples (n = 58) were collected from 3–4 plants per vineyard and brought to the laboratory to isolate and identify fungal species in symptomatic vascular tissues. Samples were debarked, washed with sterile double-distilled water (ddH2O), and sectioned to reveal vascular symptoms. Small pieces (5 mm thick) of symptomatic tissues were disinfected in 95% ethyl alcohol for 1 min, rinsed with sterile ddH2O, dried off on sterile filter papers (Whatman, Maidstone, UK) in laminar flow hood, and plated on potato dextrose agar (PDA, HiMedia, Mumbai, India), amended with streptomycin sulfate (50 μg/mL). Plates were then incubated at 25 °C in darkness until fungal colonies were observed and selected isolates were hyphal-tip-purified to obtain pure cultures maintained under the same conditions and preserved in a 40% aqueous glycerol solution at −80 °C. A preliminary identification was conducted based on colony characteristics, leading to a collection of 19 pycnidial isolates with an olivaceous color. Representative isolates were deposited in our laboratory collection and the culture collection of the Westerdijk Fungal Biodiversity Institute, Utrecht, The Netherlands (CBS) (Appendix A).

### 2.2. Genomic DNA Extraction, PCR, and Sequencing

Mycelium was collected from 1-week-old colonies grown on malt extract agar (MEA) with a sterilized spatula, avoiding taking parts of the culture medium to prevent any interference during the polymerase chain reaction (PCR). The Wizard Genomic DNA purification kit (Promega Corporation, Madison, WI, USA) was used for the genomic DNA extraction, according to the manufacturer’s protocol. Fragments of the internal transcribed spacer (ITS) region, the 28S large subunit gene of the rDNA (LSU), the 18S small subunit gene of the rDNA (SSU), the beta-tubulin gene (*b-tub*), the translation elongation factor 1-alpha (*tef1-a*), and the RNA polymerase II second-largest subunit gene (*rpb2*) were amplified using the following primer pairs: ITS5/ITS4 for ITS [51], LR0R/LR5 for LSU [52,53], NS1/NS4 for SSU [51], T1/Bt2b for *b-tub* [54,55], EF1-728F/EF2 for *tef1-a* [55,56], and 5F2/7cR for *rpb2* [57,58] (Appendix A). PCR amplifications were performed on a thermal cycler (Labcycler SensoQuest GmbH, Göttingen, Germany) in a final volume of 13 μL. The reaction mix consisted of 8.45 μL sterile ddH_2_O, 0.1 μL Taq DNA polymerase (5 U/μL BIOTAQ^TM^ DNA Polymerase, BioLine, Germany), 1.25 μL PCR buffer (10× NH_4_ reaction buffer, BioLine, Germany), 0.5 μL MgCl_2_ (50 mM, BioLine, Germany), 0.5 μL dNTP mix (10 mM, BioLine, Germany), 0.7 μL dimethyl sulfoxide (DMSO, Sigma-Aldrich, Darmstadt, Germany), 0.25 μL of each primer (10 μM), and 1 μL of DNA template. The following amplification programs were used: initial denaturation at 94 °C for 5 min, followed by 35 cycles of denaturation at 94 °C for 45 s, annealing for 45 s at 52 °C for ITS and LSU, 48 °C for SSU and *b-tub*, and 55 °C for *tef1-a* and *rpb2*, extension at 72 °C for 2 min, and a final extension at 72 °C for 8 min. For the verification of successful amplifications, electrophoreses were run on 1% (*w*/*v*) agarose gels prepared in Tris-acetate-EDTA buffer. PCR products were stained with GelRed Nucleic Acid gel stain (Biotium, Inc., Hayward, CA, USA).

For each PCR product (locus), two separate sequence reactions were performed, using the forward and reverse PCR primers, respectively, as indicated above. The sequencing PCR program consisted of an initial denaturation at 95 °C for 1 min, followed by 30 cycles of denaturation at 95 °C for 15 s, annealing at 52 °C for 14 s, and an extension at 60 °C for 4 min. The PCR products were purified using Sephadex^®^ G-50 Fine (GE Healthcare, Sigma-Aldrich, Darmstadt, Germany), according to manufacturer’s procedures. Sequencing was conducted on an ABI Hitachi 3730xl DNA Analyzer (Applied Biosystems, Foster City, CA, USA) in the facilities of the Westerdijk Institute.

In addition to the Cypriot isolates, and due to the lack of available sequences of the genetic loci *b-tub*, *tef1-a*, and *rpb2* in the NCBI database, additional representative species of the genus *Kalmusia* [*K. araucariae* (CPC 37475^Τ^), *K. ebuli* (CBS 123120^ΝΤ^), *K. longispora* (CBS 582.83^Τ^), *K. sarothamni* (CBS 116474 and CBS 113833), and *K. variispora* (CBS 121517^Τ^); (Table 1) were obtained from CBS and the working collection of Pedro W. Crous (CPC), also housed at the Westerdijk Institute, and were sequenced to ensure a more robust phylogenetic analysis.

### 2.3. Phylogenetic Analyses

Evaluations of the chromatograms and consensus sequence assembly were made using the Software Geneious Prime^®^ (version 2022.0.1) [59]. All the sequences generated in this study were submitted to GenBank, and the respective accession numbers are provided in Table 1. Single-gene alignments were performed for all loci using MAFFT v. 7 (https://mafft.cbrc.jp/alignment/server/index.html (accessed on 30 April 2024)) [60] with default settings, incorporating all available ex-type sequence data for *Kalmusia* spp., plus additional data generated in this study (Table 1). Alignments were checked and refined manually if needed using MEGA v. 6.06 software [61]. Alignments were combined using the SequenceMatrix v. 1.9 program [62] into two separated multigene datasets: first, an rDNA multigene alignment based on ITS, LSU, and SSU was used to establish the position of Cyprus *V. vinifera* fungal isolates within an overview of the complete species diversity of the genus *Kalmusia*. A second, 6-loci, multigene alignment based on *tef1-a*, ITS, LSU, *rpb2*, SSU, and *b-tub* provided the final identification of the novel Cyprus isolates.

Phylogenetic analyses were based on maximum-likelihood (ML) and Bayesian (B) analyses. Maximum-likelihood trees were constructed using IQ-TREE v. 2.1.3 [63], the best evolutionary model selection was performed using the TESTNEW option of ModelFinder [64] as implemented in IQ-TREE, and branch support was estimated using 1000 replicates of ultrafast bootstrap approximation [65]. As an additional measurement of ML branch support, 1000 replicates of the non-parametric bootstrap (BS) were calculated using RAxML-HPC2 v. 8.2.12 on ACCESS, run on the CIPRES Science Gateway portal [66]. The B analyses were run on the CIPRES Science Gateway portal using MrBayes on XSEDE v. 3.2.7a [67], and evolutionary models were calculated using MrModelTest v. 2.3, with the Akaike information criterion [68,69]. Four incrementally heated MCMC chains were run for 5M generations, with the stop-rule option on. Sampling was set to every 1000 trees, and after assessing the convergence of the runs (average standard deviation of split frequencies below 0.01), 50% consensus trees and posterior probabilities (PPs) were calculated, discarding 25% of the initial samples as the burn-in fraction.

### 2.4. Morphological Characterization

Representative isolates (CBS 151327, CBS 151329, CBS 151331, and CBS 151334) from Cyprus were described based on the colony and microscopic characteristics of cultures grown on PDA, MEA (Sigma-Aldrich, Darmstadt, Germany), and oatmeal agar (OA; Sigma-Aldrich, Darmstadt, Germany) at 25 °C for 2–3 weeks. Furthermore, segments of pine needles were autoclaved twice (121 °C for 25 min, with a 24 h interval) and placed in Petri dishes containing water agar [10]. Accordingly, mycelium plugs from actively growing cultures on PDA were placed at the center of the plates. Fungal cultures were then incubated at room temperature under alternating near-ultraviolet and white fluorescent lights (Philips, TL-D 18W BLB; Amsterdam, The Netherlands), with a dominant wavelength (λ) at 365 nm and Radium, Spectacular^©^ Plus NL-Τ8 18W/840, λ_max_ = 590 nm, respectively) in a 12/12 h photoperiod, and the formation of fruiting bodies (pycnidia) on pine needles was monitored daily for 1 week.

Morphological observations of reproductive structures were made at appropriate magnifications using a Nikon ECLIPSE 80i microscope (Nikon Corp., Tokyo, Japan) and a Nikon AZ100 Multizoom microscope (Nikon, Japan), both equipped with a color digital Nikon DS-Ri2 camera (Nikon, Japan). Conidial color, shape, length, and width were recorded using the Nikon software NIS-elements D v. 4.50, and the minimum, maximum, mean, and standard deviation were calculated. Furthermore, colony morphology and color per culture medium were rated following Rayner’s charts [70], accordingly.

### 2.5. Effect of Temperature on Mycelial Growth

The same group of isolates selected for morphological characterization was used to estimate mycelial growth cardinal temperatures. Mycelial plugs (4 mm in diameter) from the margins of actively grown cultures were transferred into Petri dishes with PDA and incubated at six temperature regimes: 10 to 30 °C in increments of 5 °C and 37 °C. Furthermore, the mycelial growth rate of the same set of isolates was recorded on MEA and OA at 25 °C, respectively. Overall, two perpendicular measurements of the diameter were recorded daily for 14 days. Three replicate plates were used per isolate, and the experiment was repeated once. The optimum temperatures for mycelial growth and the maximum daily growth rate for each isolate were calculated based on regression curves of the temperature versus daily radial growth, according to Aigoun-Mouhous et al. [71].

### 2.6. Exoenzyme Production

Endophytic fungi secrete enzymes that promote the degradation of structural macromolecules of plant cell walls, facilitating infection and colonization [1,3]. The secretion of cellulase, pectinase, and laccase of six *K. variispora* isolates (CBS 151325, CBS 151327, CBS 151329, CBS 151330, CBS 151331, and CBS 151334) were qualitatively examined on culture media that consisted of 200 mL M9 minimal salts, 5× concentrate (56.4 g per liter of ddH_2_O), 10 g of carboxymethylcellulose, 1.2 g yeast extract, 20 mL glucose (20%), 1 mL MgSO_4_ (1M), 100 μL CaCl_2_ (1M), and 15 g agar for cellulase [72], and 200 mL M9 minimal salts, 5× concentrate, 5 g pectin, 1 mL MgSO_4_ (1M), 100 μL CaCl_2_ (1M), and 15 g agar for pectinase (pectin was UV-sterilized and mixed with the sterile medium following slight heating until melting) [73,74]. PDA that was amended with guaiacol (0.01% *w*/*v*) was used for laccase detection [29]. Mycelial plugs (4 mm in diameter) were cut from the margins of actively growing cultures on PDA and placed in the center of the respective media-amended Petri dishes. Cultures were incubated at 25 °C in darkness for 10 days, and the production of the examined exoenzymes was qualitatively estimated. More specifically, for the visualization of cellulase and pectinase production, cultures were flooded with Gram’s iodine (Merck, Darmstadt, Germany) for 5 min [29,75,76]. After the cultures were rinsed with dH_2_O, the presence of distinct clearing zones around the colonies indicated enzyme production. The formation of a red-brownish zone around the developed colony on the guaiacol-amended medium was indicative of laccase production [77].

### 2.7. Pathogenicity

In May 2020, the pathogenicity of 6 isolates (CBS 151325, CBS 151327, CBS 151329, CBS 151330, CBS 151331, and CBS 151334) was evaluated on potted plants under field conditions. Two-year-old plants of the local wine grape cultivar Mavro were grown in 10 L pots filled with potting mix (Miskaar, Lambrou Agro Ltd., Limassol, Cyprus) and amended with a slow-release fertilizer (Itapollina 12-5-15 SK, Lambrou Agro Ltd., Limassol, Cyprus). Plants were aseptically wounded between the third and fourth node from the base via a cork borer. Mycelial plugs of 4 mm in diameter from 7-day-old cultures on PDA were placed in the wound, coated with Vaseline (Unilever PMT Ltd., Nicosia, Cyprus), and wrapped with Parafilm (Sigma Aldrich, Darmstadt, Germany) to avoid any contamination by any microorganism and dehydration. In addition, plants were also inoculated with sterile PDA plugs to serve as negative controls. Five plants were used per treatment in a completely randomized design, and the experiment was repeated once. Plants were placed under a shading net for sun protection and were drip-irrigated 1–2 times per week (for 30 min and 0.5 L/h) according to their need. The plants remained outdoors for 12 months and were routinely inspected for foliar symptoms. Subsequently, the inoculated stems were excised and transferred to the laboratory for further analysis. Initially, the bark was scraped off, and the length of wood discoloration was measured in both directions from the inoculation point. The canes were washed with an aqueous solution of 5% sodium hypochlorite for 2 min and then rinsed twice with sterile ddH_2_O. Subsequently, 4–6 small pieces (ca. 5 mm thick) that presented discoloration were cut from each inoculated stem. Wood segments were disinfected in 95% ethanol for 1 min and then placed on sterilized Whatman filter papers to dry in the laminar flow hood. Afterward, they were placed in PDA plates amended with streptomycin (50 mg/L) and incubated at 25 °C in the dark for 1–2 weeks. Re-isolated colonies were identified based on their morphology.

### 2.8. Statistical Analyses

Regression curves were fitted over different temperature treatments versus the mycelial growth rate for each isolate. Data were analyzed using the Kruskal–Wallis test (non-parametric ANOVA). The optimum growth temperature was also calculated according to the developed equation per isolate. Data for wood discoloration were tested for normality and homogeneity of variance using Shapiro–Wilk’s and Bartlett’s tests, respectively. ANOVA was performed to evaluate differences in the length of wood discoloration between the different fungal treatments. Since variances were homogeneous, the data were combined, and the means were compared with Tukey’s honestly significance (HSD) test at 5% of significance. All the statistical analyses were performed using SPSS (v. 25, IBM Corporation, New York, NY, USA).

## 3. Results

### 3.1. Phylogenetic Analyses

Phylogenetic analyses were performed using ML and B methods to delimit the *Kalmusia* isolates from Cyprus amongst the known diversity in the genus *Kalmusia*. The composition of the different alignments and analyses is shown in Appendix A. The first analysis of rDNA sequences encompassed 25 isolates representing nine *Kalmusia* spp., plus three outgroup taxa (*Neokalmusia brevispora* CBS 120248, *Paraconiothyrium estuarinum* CBS 109850^T^, and *Paraconiothyrium hakeae* CBS 142521) (Appendix A). Phylogenetic analyses showed that the *Kalmusia* isolates from Cyprus clustered within two non-clearly delimited sister groups, including the ex-type strains of *K. variispora* (CBS 121517^Τ^).

To establish an unequivocal phylogenetic position of the *Kalmusia* isolates from Cyprus and *K. variispora*, a second analysis was performed based on a six-loci combined alignment (*tef1-a*, ITS, LSU, *rpb2*, SSU, and *b-tub*), which included sequences from 21 isolates, representing five *Kalmusia* spp. plus the three outgroup taxa, as indicated above (Figure 1). An initial assessment of the dataset using ML and B methods showed that the analyses were quite sensitive to missing data, largely affecting the three topologies and node support values; hence, taxa lacking sequence data for two or more loci and for which ex-type strains were not available for the generation of additional sequence data (*K. cordylines* ZHKUCC 21-0092^Τ^, *K. erioi* MFLUCC 18-0832^Τ^, *K. italica* MFLUCC 13-0066^Τ^, and *K. spartii* MFLUCC 14-0560^Τ^) were excluded from the analyses. The analyses showed that the *Kalmusia* isolates from Cyprus confidently resolved within a fully supported clade with the ex-type strain of *K. variispora* (CBS 121517^Τ^). With the exceptions of *K. sarothamni* (supported by BS only) and the outgroup taxon *N. brevispora* (not supported by any method), all additional species were confidently resolved in this analysis.

### 3.2. Morphological Characterization

Cultures on PDA were felty and flat, pale olivaceous with white aerial mycelium at the center and a smooth white margin (Figure 2). On their reverse side, colonies were grey olivaceous and smoke grey to olivaceous buff on the center with a white margin. On MEA, the colonies were felty and flat, white with immersed pale grey mycelium in the center with a smooth margin, while their reverse side was pale luteous to pale buff. Colonies on OA were white with aerial, cottony mycelium, immersed glaucous grey to pale-grey mycelium at the center, with a colorless to white margin. On the reverse side, the colonies were glaucous grey to pale olivaceous buff.

Conidiophores were 1–4-septate, acropleurogenous, simple, or branched at the base, ranging 6.0–11.7 μm × 2.4–4.3 μm (av. 8.8 ± 1.3 × 3.2 ± 0.5 μm). Conidiogenous cells were phialidic, obclavate, narrowed toward the tip, hyaline, integrated, terminal, and intercalary. Conidia were aseptate, ellipsoid to cylindrical, commonly straight, rarely curved with round tips, hyaline, thin-walled, containing 1–3 min oil droplets, ranging from 3.2 to 4.1 μm × 1.3 to 1.8 μm (av. 3.6 ± 0.2 × 1.5 ± 0.1 μm), with an average L/W ratio of 2.4 (Figure 2). Pycnidia were spherical or irregular, dark brown to black, ranging from 150 to 393 μm in diameter.

### 3.3. Effect of Temperature on Mycelial Growth

All *K. variispora* isolates from Cyprus grew at all tested temperatures except at 37 °C. The analyses of variance indicated no significant differences (*p* < 0.05) in the mycelial growth among experiments; thus, the data were combined. The correlation of mycelial growth to growth temperature was best described by a quadratic polynomial response model (y = aT^2^ + bT + c). The optimum temperature for radial growth and the maximum daily radial growth were calculated in a fitted equation for each isolate. The coefficients of determination (R^2^) ranged from 0.89 to 0.93, and based on the adjusted models derived per isolate, the optimum temperatures of mycelial growth on PDA ranged from 22.18 to 22.92 °C (mean = 22.5 °C) (Table 2). The maximum mycelial growth rates were not significantly different, ranging from 3.92 to 4.36 mm/day (mean = 4.18 mm/day) (Table 2).

Mycelial growth rates of *K. variispora* isolates were also estimated on MEA, OA, and PDA at 25 °C (Appendix A). The average growth rate on MEA ranged from 3.13 to 3.57 mm/day (mean = 3.40 mm/day). The highest average growth rate for all *K. variispora* isolates was evident on OA, ranging from 4.06 to 4.24 mm/day (mean = 4.14 mm/day), compared to ΜΕA and PDA (Appendix A). Overall, all tested isolates exhibited the highest growth rate on OA and the lowest on MEA (Appendix A).

### 3.4. Exoenzyme Production

Digestive exoenzyme production revealed differences between the tested isolates in this study (Figure 3). All isolates were positive in cellulase production, indicated by a distinct, clear, and prominent zone of clearance around the developed colonies with bluish-black coloration of the medium (Figure 3A–F). Equally large clearance zones were also evident for all isolates on the pectin agar medium, indicating pectinolytic activity (Figure 3G–L). On the other hand, *K. variispora* isolates showed differential laccase activity (Figure 3M–R), ranging from wide to faint reddish-brown halos around the colony, with the absence of a zone in the case of the isolate CBS 151329 (Figure 3P).

### 3.5. Pathogenicity

Twelve months post-inoculation, all tested *K. variispora* isolates were found pathogenic to 2-year-old cv. Mavro potted grapevines. Light to dark-brown wood discoloration and lesions of varying levels, ranging from 12.3 to 4.0 cm long, developed upward and downward from the inoculated point of the canes (Figure 4). Isolates CBS 151325, CBS 151327, and CBS 151331 were significantly more aggressive (12.3 ± 3.6 cm to 7 ± 1.5 cm) compared to CBS 151329, CBS 151330, and CBS 151334 (4 ± 1 to 6.3 ± 1.6 cm) (*p* < 0.0001) (Figure 5). No symptoms developed on non-fungal-inoculated plants (Figure 4). Successful re-isolations were positive from all the fungal-inoculated plants, with recovery percentages ranging from 45 to 84%. Retrieved isolates were similar to the ones used in grapevine inoculations based on morphology (cultural and conidial characteristics), while no fungal isolates were obtained from the negative-control plants.

## 4. Discussion

This study provides the first evidence of *K. variispora* associated with GTDs in Cyprus, offering a comprehensive cultural, morphological, and phylogenetic description of this plant-pathogenic species. Despite the widespread impact of GTDs, research on their etiology in Cyprus [10,78] has been limited, making this study a valuable contribution to the field. *Kalmusia variispora* isolates were collected from grapevines of wine and table-grape cultivars across Cyprus, exhibiting typical GTD symptoms such as cankers, dead cordons, and spurs. These symptoms are commonly associated with vines affected by Botryosphaeria dieback, Eutypa dieback, or Phomopsis dieback.

*Kalmusia variispora* (formerly *Dendrothyrium variisporum*) was initially isolated from declined grapevines in Syria and *Erica carnea* (common name: Winter heath) in Switzerland [37]. However, the pathogenicity of *K. variispora* on grapevines was first confirmed in Iran [28], with subsequent reports from vineyards in North America (California and Washington) and Greece [23,47]. Additionally, Bekris et al. identified *K. variispora* as an indicator of GTD-symptomatic grapevine plants due to its significantly higher relative abundance in the symptomatic plants in Greece [13]. Beyond grapevines, *K. variispora* has been reported as a major causal agent of the apple fruit core rot in Greece and Chile [79,80] and as an endophyte of *Rosa hybrida* and *Pinus eldarica* (common names: Rose and Afghan pine, respectively) [28], while it has also been associated with oak decline in Iran, causing defoliation, vascular discoloration, and wood necrosis [45]. Furthermore, *K. variispora* has been isolated from the necrotic wood of pomegranate, although pathogenicity has not been confirmed [39]. *Kalmusia longispora*, a sister species of *K. variispora*, has also been associated with GTDs [29] and apple tree diebacks [46] in Hungary and Italy, respectively.

The presence of GTD-related species has been commonly documented from asymptomatic mother plants and mature vines [81,82,83], suggesting that they may remain latent for a period before transitioning to their pathogenic phase [84]. Among these pathogens, species in the genus *Ilyonectria* are considered weak and/or opportunistic pathogens [84]. GTDs are caused primarily by taxonomically unrelated Ascomycete fungi and, to a lesser extent, by Basidiomycete fungi [85]. Beyond the well-known GTD pathogens, new reports have identified other fungi associated with GTDs, often isolated alongside the established GTD pathogens [10,26,29]. The co-isolation of *K. variispora* in the current work with other GTD pathogens highlights the complex etiology of GTDs. Among the nineteen *K. variispora* isolates, seven were co-isolated from vine cankers with other known GTD-related pathogens in Botryosphaeriaceae, the genera *Phaeoacremonium* or *Seimatosporium*, as well as with *Phaeomoniella chlamydospora.* Those isolates were identified based on colony and morphological features, and by the sequencing of the ITS locus. Mixed fungal infections are commonly found in vineyards, supporting the complex nature of GTDs [10,13]. There is evidence that in some cases, the simultaneous presence of more than one pathogen increases or decreases the severity of plant disease [26]. Mixed infections could exacerbate disease severity and complicate management strategies.

Phylogenetic analyses revealed that all *Kalmusia* isolates from Cyprus clustered with the type strain of *K. variispora* (Figure 1), confirming their identity as *K. variispora*. Currently, the available sequences of *Kalmusia* species in the NCBI database predominately refer to the ITS, LSU, and SSU genetic loci, representing nine different species. The present work concluded that the LSU and SSU do not provide enough phylogenetically informative sites to support the identification of *Kalmusia* at the species level (Appendix A). However, the remaining loci (ITS, *b-tub*, *rpb2*, and *tef1-a*) were significantly more informative; therefore, we recommend using these for future phylogenetic studies on *Kalmusia.* In addition, we provide molecular data from the available type strains preserved at the WI, making them accessible for future research through the NCBI database. Furthermore, four representative *K. variispora* isolates we isolated from local vineyards in Cyprus shared the same morphological characters with the type strain of *K. variispora* species described by Verkley et al. [37].

Abed-Ashtiani et al. conducted a pathogenicity assay on *K*. *variispora* using green shoots of two-year-old potted grapevines in a greenhouse for a ten-week incubation period [28]. The infected plants exhibited wilting, general decline, and brown discoloration of the wood. Species pathogenicity was further confirmed in North America by Travadon et al., who inoculated woody stems of potted grapevine plants in greenhouse conditions with a conidial suspension of 10 species (*Biscogniauxia mediterranea*, *Cadophora columbiana*, *Cytospora viticola*, *Cytospora yakimana*, *Eutypella citricola*, *Flammulina filiformis*, *K. variispora*, *Phaeomoniella chlamydospora*, and *Thyrostroma* sp.) found in association with GTDs in the states of California and Washington [47]. Testempasis et al. assessed the pathogenicity of *K. variispora* on two-year-old grapevine canes in a 15-year-old vineyard for a six-month incubation period, where inoculated canes exhibited severe wood discoloration [23]. All *K. variispora* isolates from the present study also caused light to dark-brown wood discoloration symptoms, consistent with the symptoms observed in the field-collected samples and were successfully re-isolated from the inoculated wood. The above-mentioned findings confirmed the presence of *K. variispora* in the GTD-associated microbiome from countries in different continents and its ability to infect and colonize the grapevine wood, suggesting that it is a pathogen associated with grapevine trunk diseases worldwide.

The plant cell wall also serves as a structural barrier element against microbial infections that successful pathogenic microorganisms must overcome or degrade to invade their hosts [86]. To achieve that, pathogens need to secrete enzymes that degrade cell wall components such as cellulose, hemicellulose, pectin, and lignin, particularly in woody tissues [48]. The secretion of these degrading enzymes is a key pathogenicity factor that contributes to the pathogen’s aggressiveness [87]. Karácsony et al. evaluated *K. longispora*, another grapevine pathogen, and found that it secretes cellulase, pectinase, and laccase, with laccase likely playing a significant role in its ability to cause vascular necrosis in grapevines [29]. More specifically, the more pathogenic isolates CBS 151325, CBS 151327, and CBS 151331 (Figure 5) exhibited pronounced laccase production (Figure 3) compared to CBS 151334, CBS 151330, and CBS 151329, which were less aggressive and faint, or without laccase producers. Thus, laccase production by *K. variispora* could be qualitatively related to pathogenicity in terms of wood discoloration and lesion length on lignified grapevine canes. In the present study, the observed variability in ligninase production, where one isolate was not able to produce lignin-degrading enzymes, provides insights into the diverse enzymatic strategies of *K. variispora*. The consistent production of enzymes that degrade cellulose and pectin across all *K. variispora* isolates from grapevines in Cyprus emphasizes the importance of these enzymes in the pathogenicity of the species. Furthermore, *K. variispora* has been shown to produce a wide array of phytotoxic metabolites [88].

Herein, we showed the ability of *K. variispora* to cause vascular discoloration on grapevines, as well as the possible importance of laccases in the development of its symptoms. This study contributes to the growing body of knowledge on GTD pathogens in Mediterranean vineyards. To the best of our knowledge, this is the first report of *K. variispora* in grapevines in Cyprus.

## Figures and Tables

**Figure 1 pathogens-14-00428-f001:**
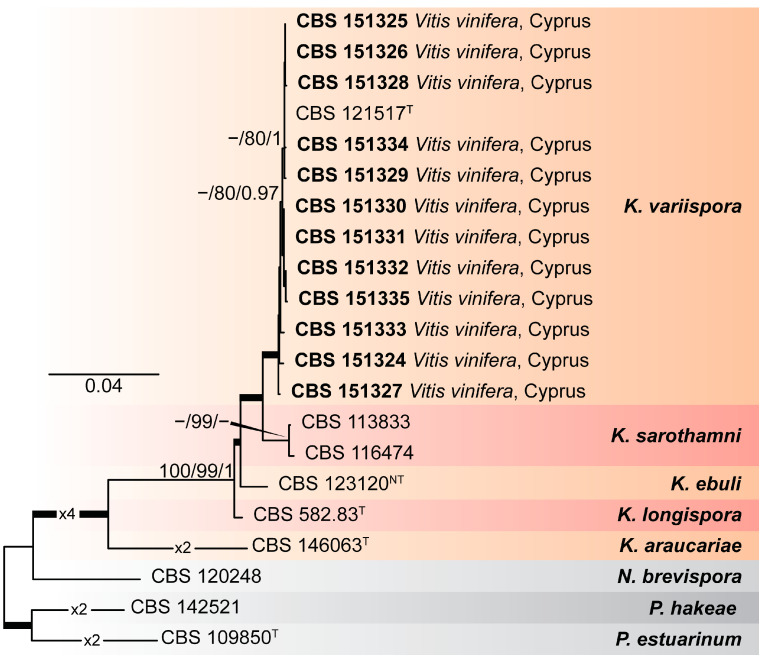
Maximum-likelihood (IQ-TREE-ML) consensus tree inferred from the combined 6-loci dataset (*tef1-a*, ITS, LSU, *rpb2*, SSU, and *b-tub*) of representative species of the genus *Kalmusia*. Numbers at the nodes indicate support values (IQ-TREE Uboot, RAxML-BS, and B-PP) above 70% (Uboot and BS) and 0.95 (B). Thickened branches indicate full support (Uboot and BS = 100%, and PP = 1). The scale bar indicates expected changes per site. The tree is rooted to *Neokalmusia brevispora* CBS 120248, *Paraconiothyrium estuarinum* CBS 109850^T^, and *Paraconiothyrium hakeae* CBS 142521. Ex-neotype and ex-type strains are indicated with ^NT^ and ^T^, respectively.

**Figure 2 pathogens-14-00428-f002:**
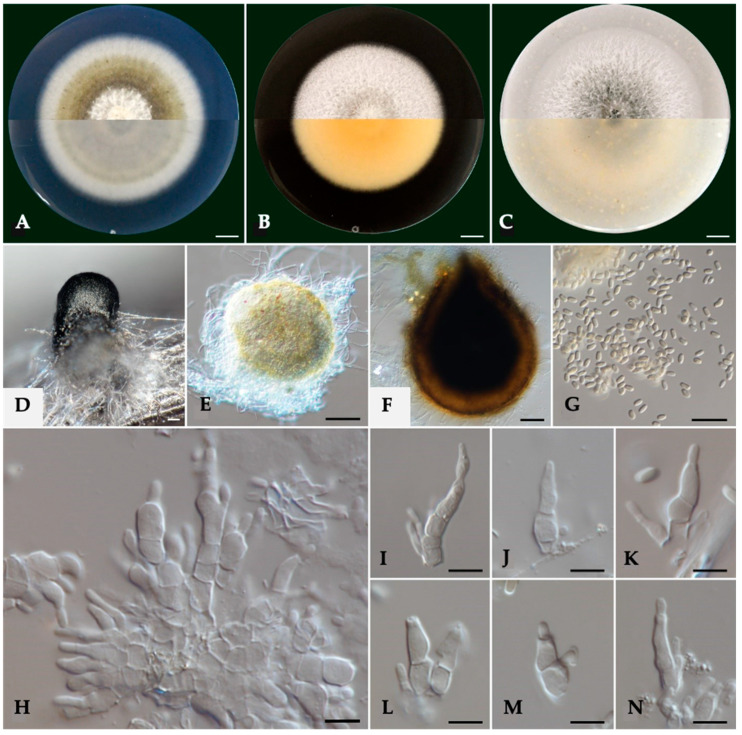
Morphology of *Kalmusia variispora* isolate CBS 151327. (**A**–**C**) Colonies cultured on PDA, MEA, and OA, respectively, inoculated at 25 °C in the darkness for 14 days (top: above and bottom: reverse of the plate). (**D**) Pycnidium on a pine needle; (**E**,**F**) pycnidia from 7 day-old and 11-day-old colonies, respectively; (**G**) conidia; (**H**) group of conidiogenous cells and conidiophores; (**I**–**K**) conidiogenous cells and conidiophores; and (**L**–**N**) different conidium growing stages. Scale bars: 10 mm (**A**–**C**), 50 μm (**D**–**F**), 10 μm (**G**), and 5 μm (**H**–**N**).

**Figure 3 pathogens-14-00428-f003:**
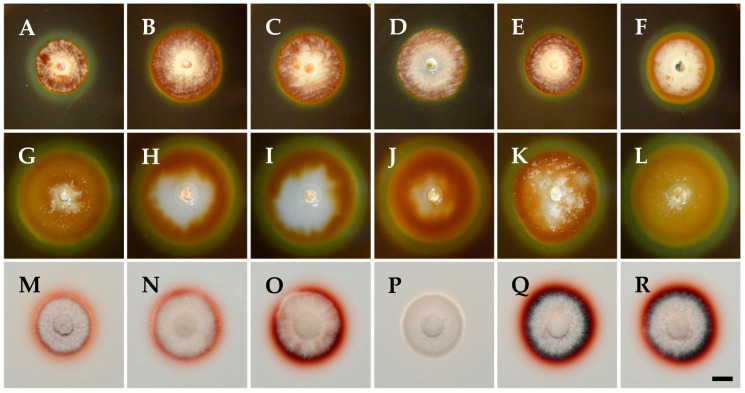
Exoenzyme production of *Kalmusia variispora* isolates CBS 151334 (**A**,**G**,**M**), CBS 151330 (**B**,**H**,**N**), CBS 151331 (**C**,**I**,**O**), CBS 151329 (**D**,**J**,**P**), CBS 151327 (**E**,**K**,**Q**), and CBS 151325 (**F**,**L**,**R**) growing on indicative media for cellulase (**A**–**F**), pectinase (**G**–**L**), and laccase (**M**–**R**) production. Photographs were taken after 10 days of incubation at 25 °C in darkness. Scale bar: 10 mm.

**Figure 4 pathogens-14-00428-f004:**
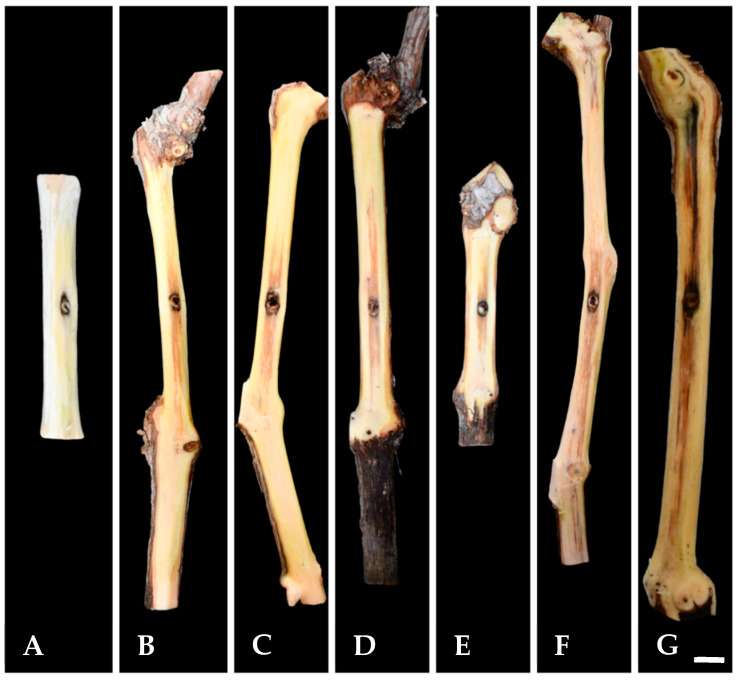
Pathogenicity of *Kalmusia variispora* isolates collected from vineyards in Cyprus, on 2-year-old potted vines (cv. Mavro). Wood discolorations and lesions were caused on lignified canes by isolates CBS 151334 (**B**), CBS 151330 (**C**), CBS 151331 (**D**), CBS 151329 (**E**), CBS 151327 (**F**), and CBS 151325 (**G**) 12 months post-inoculation. Mock-inoculated canes (**A**) did not exhibit symptoms. Scale bar: 10 mm.

**Figure 5 pathogens-14-00428-f005:**
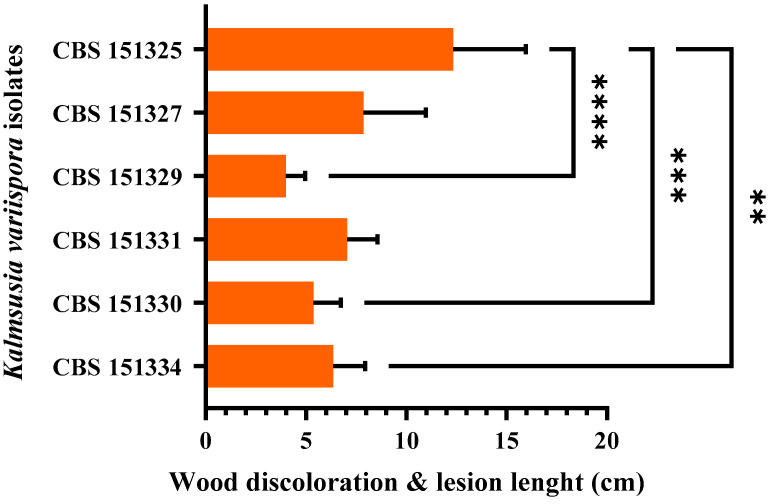
Mean lesion and wood discoloration lengths caused at 12 months post-inoculation by *Kalmusia variispora* isolates on 2-year-old vines (cv. Mavro) in pathogenicity assays under field conditions. Each bar represents an individual tested isolate and vertical error bars indicate the corresponding standard deviation. Asterisks (****, ***, and **) indicate statistically significant differences (*p* <0.0001, *p* < 0.001, and *p* < 0.01), following the analysis of variance and Tukey’s mean separation test procedures.

**Table 1 pathogens-14-00428-t001:** Fungal species used for phylogenetic analyses in the current study.

Species	Strain ^a,b^	Country	Host	GenBank Accession Number ^c^					
				ITS	LSU	SSU	*b-tub*	*rpb2*	*tef1-a*
*Kalmusia araucariae*	CPC 37475^T^	USA	*Araucaria bidwillii*	MT223805	**PP664292**	**PP667340**	-	**PP715462**	**PP715503**
*K. cordylines*	ZHKUCC 21-0092^T^	China	*Cordyline fruticosa*	OL352082	OL818333	OL818335	-	-	-
*K. ebuli*	CBS 123120^NT^	France	*Populus tremula*	**PP667315**	**PP664289**	**PP667335**	**PP715468**	**PP715447**	**PP715488**
*K. erioi*	MFLUCC 18-0832^T^	Thailand	*-*	MN473058	MN473052	MN473046	MN481603	-	-
*K. italica*	MFLUCC 13-0066^T^	Italy	*Spartium junceum*	KP325440	KP325441	KP325442	-	-	-
*K. longispora*	CBS 582.83^T^	Canada	*Arceuthobium pusillum*	JX496097	JX496210	**PP667338**	**PP715481**	**PP715460**	**PP715501**
*K. sarothamni*	CBS 116474	-	*-*	**PP667313**	**PP664288**	**PP667332**	**PP715466**	**PP715444**	**PP715485**
*K. sarothamni*	CBS 113833	-	*-*	**PP667311**	**PP664287**	**PP667331**	**PP715463**	**PP715442**	**PP715483**
*K. spartii*	MFLUCC 14-0560^T^	Italy	*S. junceum*	KP744441	KP744487	KP753953	-	-	-
*K. variispora*	CBS 121517^T^	Syria	*Vitis vinifera*	**PP667314**	JX496143	**PP667334**	**PP715467**	**PP715446**	**PP715487**
*K. variispora*	CBS 151324	Cyprus	*V. vinifera*	**PP667316**	**PP664290**	**PP667336**	**PP715469**	**PP715448**	**PP715489**
*K. variispora*	CBS 151325	Cyprus	*V. vinifera*	**MZ312148**	**MZ312183**	**MZ312321**	**PP715470**	**PP715449**	**PP715490**
*K. variispora*	CBS 151326	Cyprus	*V. vinifera*	**MZ312149**	**MZ312184**	**MZ312322**	**PP715471**	**PP715450**	**PP715491**
*K. variispora*	CBS 151327	Cyprus	*V. vinifera*	**MZ312138**	**MZ312173**	**MZ312311**	**PP715472**	**PP715451**	**PP715492**
*K. variispora*	CBS 151328	Cyprus	*V. vinifera*	**PP667317**	**PP664291**	**PP667337**	**PP715473**	**PP715452**	**PP715493**
*K. variispora*	CBS 151329	Cyprus	*V. vinifera*	**MZ312140**	**MZ312175**	**MZ312313**	**PP715474**	**PP715453**	**PP715494**
*K. variispora*	CBS 151330	Cyprus	*V. vinifera*	**MZ312141**	**MZ312176**	**MZ312314**	**PP715475**	**PP715454**	**PP715495**
*K. variispora*	CBS 151331	Cyprus	*V. vinifera*	**MZ312143**	**MZ312178**	**MZ312316**	**PP715476**	**PP715455**	**PP715496**
*K. variispora*	CBS 151332	Cyprus	*V. vinifera*	**MZ312139**	**MZ312174**	**MZ312312**	**PP715477**	**PP715456**	**PP715497**
*K. variispora*	CBS 151333	Cyprus	*V. vinifera*	**MZ312144**	**MZ312179**	**MZ312317**	**PP715478**	**PP715457**	**PP715498**
*K. variispora*	CBS 151334	Cyprus	*V. vinifera*	**MZ312142**	**MZ312177**	**MZ312315**	**PP715479**	**PP715458**	**PP715499**
*K. variispora*	CBS 151335	Cyprus	*V. vinifera*	**MZ312151**	**MZ312186**	**MZ312324**	**PP715480**	**PP715459**	**PP715500**
*Neokalmusia brevispora*	CBS 120248	Japan	*Sasa* sp.	MH863078	JX681110	**PP667333**	**PP715464**	**PP715445**	**PP715486**
*Paraconiothyrium hakae*	CBS 142521	Australia	*Hakea* sp.	KY979754	KY979809	-	KY979920	KY979847	KY979892
*P. estuarinum*	CBS 109850^T^	Brazil	estuarine sediment	MH862842	MH874432	AY642522	JX496355	LT854937	-

^a^ CBS: Culture collection of the Westerdijk Fungal Biodiversity Institute, Utrecht, The Netherlands; CPC: Culture collection of Pedro W. Crous, housed at the Westerdijk Institute; MFLUCC: Mae Fah Luang University Culture Collection Center of Excellence in Fungal Research, Mae Fah Luang University, Chiang Rai, Thailand; NBRC: Biological Resource Center, National Institute of Technology and Evaluation, Chiba, Japan. ^b^ Status of the isolates = ^T^: ex-type strain; ^NT^: neotype strain. ^c^ GenBank accession numbers for the sequences of six loci: ribosomal DNA (rDNA) internal transcribed spacer region (ITS), rDNA large subunit (LSU), rDNA small subunit (SSU), b-tubulin (*b-tub*), RNA polymerase II gene (*rpb2*), and translation elongation factor 1-a (*tef1-α*). All Cyprus isolates (CBS 151324-151335) were collected in the present study. Sequences generated in the current work are indicated in bold.

**Table 2 pathogens-14-00428-t002:** Temperature–mycelial growth relationship for four *Kalmusia variispora* isolates collected from vineyards in Cyprus.

Isolate	Adjusted Model ^x^				Optimum Temperature (°C) ^y^	Growth Rate (mm/Day) ^z^
	R^2^	*a*	*b*	*c*		
CBS 151327	0.89	−0.017	0.7850	−47.854	22.18 a	3.92
CBS 151329	0.90	−0.019	0.8649	−54.839	22.64 a	4.31
CBS 151331	0.90	−0.020	0.9305	−62.999	22.92 a	4.36
CBS 151334	0.93	−0.018	0.8052	−483.334	22.24 a	4.12
Mean					22.5	4.18

^x^ Data are the mean of six replicates per isolate. Means with the same letter, within a column, are not significantly (*p* = 0.05) different according to Kruskal–Wallis and the Dunn’s test for multiple comparisons; mycelial growth of PDA at 10, 15, 20, 25, 30, and 37 °C was adjusted to a quadratic model: y = aT^2^ + bT + c, with y = mycelial growth (mm/day); a, b, c = regression coefficients and R^2^ = coefficient of determination. ^y^ Optimum temperatures per isolate were estimated by the adjusted model. ^z^ Maximum growth rate per isolate was estimated by the adjusted model.

## Data Availability

The datasets generated during and/or analyzed during the current study are available from the corresponding author on reasonable request.

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
