# Peer review of "Kalmusia variispora (Didymosphaeriaceae, Dothideomycetes) Associated with the Grapevine Trunk Disease Complex in Cyprus"

_pathogens, 2025, doi:10.3390/pathogens14050428_

Round 1

Reviewer 1 Report

Comments and Suggestions for Authors

The paper, devoted to clarification of role of a particular phytopathogenic microorganism to the grapevine trunk diseases, fairly corresponds to the journal scope. It is performed using a robust set of molecular markers, convenient in modern fungal taxonomy, accompanied with excellent microscopic observations. The study is further supplemented with vivid enzymatic profiles and examination of pathology to establish importance of the fungus. The work is well designed and structured. The manuscript is written in literate English. The illustrations are of high quality. The introduction extensively covers the topic of the research and substantiates the goal. Methods are provided in a meticulous way and results are clear and well-presented. The Discussion is sufficient and conclusions are convincing. This is a nice example of a taxonomic and phytopathological  

Few minor corrections/comments are provided below. Minor revision seems to be a proper solution for this submission.

Title: I would recommend expanding title to indicate taxonomy of the fungus

Abstract (and possibly elsewhere): I would avoid term "pathogenic association" as an obscure one. Pathogenic are the fungi, not the associations.

Line 34: I would argue that plant diseases emerged long before the humankind started to cultivate crop, if only specific conditions favor development of diseases which are absent in nature and these diseases develop specifically in culture

Line 44: please check (and use uniformly) whether the Journal's guides imply italics in Latin taxa epithets beyond genus and species rank

Lines 172-174: I guess such details are not necessary

Line 209: observations are made, performed etc, not determined 

Line 251: can you briefly justify the use of six particular isolates for the pathogenicity tests?

Table 2: can be moved to supplementary files

Fig1 vs Fig2: I do not see the rationale in providing reconstruction as in Fig1 since Fig2 shows a more robust phylogeny

Line 406: confirmed with those - sounds obscure

L433 etc: please provide trivial plant names, if applicable

Author Response

Dear Reviewer 1,

We would like to acknowledge your comments that helped us to shape our manuscript in the best possible way. All suggestions were valuable, and we addressed them to the best we could. 

Comments 1 Title: I would recommend expanding title to indicate taxonomy of the fungus

Response 1: We have expanded the title in the revised manuscript accordingly, “Kalmusia variispora (Didymosphaeriaceae, Dothideomycetes) associated with the grapevine trunk disease complex in Cyprus”.

Comments 2 Abstract (and possibly elsewhere): I would avoid term "pathogenic association" as an obscure one. Pathogenic are the fungi, not the associations.

Response: We agree with the comment thus we have already made the appropriate corrections in the revised manuscript. More specifically, in the lines 22-23 of the abstract “The present study confirmed the association of K. variispora with grapevine as a pathogen…”, as well as, in the line 78 of the Introduction in the revised manuscript “(b) to evaluate its potential as pathogen under local conditions”.

Comments 3 Line 34: I would argue that plant diseases emerged long before the humankind started to cultivate crop, if only specific conditions favor development of diseases which are absent in nature and these diseases develop specifically in culture

Response: I agree with your comment that plant diseases mostly emerged despite the onset of crop cultivation by humans. In lines 33-34, we wrote “Although GTDs are considered to be as old as Vitis vinifera cultivation (Larignon and Dubos 1997, Mugnai et al. 1999)…” to emphasize the inextricable link between the Grapevine Trunk Diseases and grapevines. Nevertheless, in the revised manuscript, we rephrased the sentence as: “Although GTDs are inextricably associated with Vitis vinifera (Larignon and Dubos 1997, Mugnai et al. (1999) …”.

Comments 4 Line 44: please check (and use uniformly) whether the Journal's guides imply italics in Latin taxa epithets beyond genus and species rank

Response: We considered writing the family names in italics; however, in the revised manuscript, we have reverted family names in plain text font. The changes can be found in lines 41, 52, and 439 of the revised manuscript.

Comments 5 Lines 172-174: I guess such details are not necessary

Response: We agree with the suggestion, and we have removed the sentence “A 173 bp indel in Neokalmusia brevispora (CBS 120248), and a 178 bp unalignable region were removed from the ITS and tef1-a alignments, respectively, to ensure unambiguous alignment of all the included sequences” from the revised manuscript.

Comments 6 Line 209: observations are made, performed, etc., not determined

Response: We agree with the comment; thus, we have replaced “determined” with “made” (line 192 in the revised manuscript).

Comments 7 Line 251: Can you briefly justify the use of six particular isolates for the pathogenicity tests?

Response: The isolates used in the pathogenicity test (lines 231-232 in the revised manuscript) were selected to represent different sampling areas (provinces of Paphos, Limassol, and Nicosia) and V. vinifera cultivars, as shown in the Supplementary Table S1.

Comments 8 Table 2: can be moved to supplementary files

Response: We have moved “Table 2” to the supplementary files (now entitled Supplementary Table S3). Subsequently, we have renumbered the Tables in the revised manuscript.

Comments 9 Fig1 vs Fig2: I do not see the rationale in providing reconstruction as in Fig1 since Fig2 shows a more robust phylogeny

Response: Currently, the available sequences of Kalmusia species in the NCBI database predominately refer to the ITS, LSU and SSU genetic loci, that according to our work do not provide a robust phylogenetic analysis of the genus (Supplementary Figure S1), compared to that inferred from the combination of 6-loci (tef1-a, ITS, LSU, rpb2, SSU, and b-tub; as shown in Figure 2). The reason we included the two phylograms was to demonstrate the improvement of the analysis with the incorporation of new loci. However, we agree with your comment, which is why we moved “Figure 1” to the supplementary files of the revised manuscript (now entitled Supplementary Figure S1).

Comments 10 Line 406: confirmed with those - sounds obscure

Response: We agree with the comment, thus, we replaced the sentence “Retrieved isolates were confirmed with those used in grapevine inoculations…” with “Retrieved isolates were similar to the ones used in grapevine inoculations…” (lines 373-374 in the revised manuscript).

Comments 11 L433 etc: please provide trivial plant names, if applicable

Response: In the first version of the submitted manuscript, we used the scientific names of the plant species mentioned in the respective scientific papers. However, in the revised manuscript, we included the common names of the plant species Erica carnea (Winter heath), Rosa hybrida, and Pinus eldarica (Rose and Afghan pine, respectively), in lines 416 and 423-424, respectively.

Reviewer 2 Report

Comments and Suggestions for Authors

The current paper devoted to investigation of the grapevine trunk disease complex in Cyprus and it association with Kalmusia variispora.

Authors tested 19 nisolates and have found that they have cellulase, pectinase, and laccase production.

The quiestions authors asked is very important from economical and scientifical points of view.  

One othe main funding is the possible importance of laccases in the development of Infection  symptoms..

The paper is well. Written

Some minor corrections are required. Maybe in final version authors re-numbering citations according to MDPI rule.

Minor points:

Line 105: must be 25°C.

Line 216: „Effect of temperature on mycelial growth“ – the subtitle look as results. Maybe „studying the temperature varíation“ will be better?

Lines 248, 297 : double point.

 Figure 3: not all images have a scale bar.

Fig 4: very informative, but scale bar will be usefull.

Fig 5: scale bar

Line 500: „The plant cell wall serves as a dynamic barrier against microbial infections“ : the plant cell wall is rather s structural elements, plasma membrane is a real barrier.

Author Response

Dear Reviewer 2,

We would like to acknowledge your comments that helped us to shape our manuscript in the best possible way. All suggestions were valuable, and we addressed them to the best we could. 

Comment 1 Maybe in final version authors re-numbering citations according to MDPI rule.

Response 1: Thank you for pointing this out. We agree with this comment, and we have renumbered the citations of the manuscript according to the MDPI format.

Comment 2 Line 105: must be 25°C.

Response 2: We have made the suggested correction in line 92 of the revised manuscript.

Comment 3 Line 216: “Effect of temperature on mycelial growth “- the subtitle looks like results. Maybe “studying the temperature variation“ will be better?

Response 3: The scope of this objective was to study the effect of a broad range of temperatures (10 to 30°C at 5°C increments and at 37°C) to the mycelial growth of Kalmusia variispora, thus we believe the present subtitle supports the above-mentioned scope and we would like to keep it the same.

Comment 4 Lines 248, 297: double point.

Response 4: We agree with the comment, and we removed the extra points in the revised manuscript.

Comment 5 Figure 3: not all images have a scale bar.

Response 5: We agree with the comment, and we have added the scale bars that correspond to the sub-figures A, B, C, and D, as well as the appropriate description of the scale size in the footnote of Figure 2 in the revised manuscript (lines 323-324).

Comment 6 Fig 4: very informative, but scale bar will be usefull.

Response 6: We agree with the comment, and we have added the scale bars, as well as the appropriate description of the scale size in the respective footnote of Figure 3 (line 363) in the revised manuscript.

Comment 7 Fig 5: scale bar

Response 7: The scale bar has been added, as well as the appropriate description of the scale size in the respective footnote of Figure 4 (lines 405-406) in the revised manuscript.

Comment 8 Line 500: “The plant cell wall serves as a dynamic barrier against microbial infections“: the plant cell wall is rather s structural elements; plasma membrane is a real barrier.

Response 8: We appreciate the valuable suggestion and agree with it. Therefore, we have rephrased the sentence accordingly as follows: “The plant cell wall serves also as a structural barrier element against microbial infections that successful pathogenic microorganisms…¨, as can be seen in lines 476-478 of the revised manuscript.

Reviewer 3 Report

Comments and Suggestions for Authors

The manuscript describes well-organized study with the appropriate choice of methods and approaches for determining pathogens associated with the GTD disease of grape. The text is well-structured, Introduction section quite completely describes the problem. The purpose of the study is clearly formulated, the methods and approaches are described in details. Tables and figures presented in the Results section are informative, and conclusions are supported by the obtained data.

In my opinion, this manuscript corresponds to the scope of the journal and is of good quality (I have only several minor comments - see below).

The language quality is good, no editing is needed.

Line 95: how many samples were collected for this study? Please, add this information.

Line 109: just a small clarification - did you obtain 19 isolated in total from all collected samples? How many of them were considered as representative ones (which criteria did you use to choose them?) and how many of them were sequenced? This information will help to evaluate the total volume of the performed work within this study.

Table 1: it would be good to adjust the column width or the font size in such a way that all symbols of accession numbers remained within one row. Table title: I suggest “Fungal species” would better suit to this table than “Fungal taxa”.

Line 201-202: please, explain the sentence about pine needles. Why did you use them to observe pycnidia formation? Is this a common method?

Line 413: I suggest there should be CBS 151331 instead of CBS 151221.

Lines 508-510: CBS 151329, which showed the lack of laccase-degrading enzymes, also showed the lowest pathogenicity, while CBS151325, which showed the maximum pathogenicity, also demonstrated a significant laccase-degrading activity. Did you make a correlation analysis between these two traits? It would be good to illustrate these lines of the Discussion section with these data.

Line 525-528: I do not find any author who did investigation itself. Please, check and correct.

References: please, check for the correspondence to the journal rules. For example, compare Ref. 1 and Ref 2 (placement of the year of publication).

Author Response

Dear Reviewer 3,

We would like to acknowledge your comments that helped us to shape our manuscript in the best possible way. All suggestions were valuable, and we addressed them to the best we could. 

Comment 1 Line 95: how many samples were collected for this study? Please, add this information.

Response 1: As mentioned in the manuscript (lines 82-87 of the revised manuscript), 15 vineyards of various grape cultivars were sampled from various areas in the provinces of Paphos (n=7), Limassol (n=6), and Nicosia (n=2) in the country of Cyprus (Supplementary Table S1). Wood samples were collected from 3–4 plants per vineyard. In total, 58 wood samples were collected (line 85 in the revised manuscript) from vines exhibiting typical symptoms caused by grapevine trunk diseases.

Comment 2 Line 109: just a small clarification - did you obtain 19 isolated in total from all collected samples? How many of them were considered as representative ones (which criteria did you use to choose them?) and how many of them were sequenced? This information will help to evaluate the total volume of the performed work within this study.

Response 2: Overall, 84 fungal strains were isolated from the sampled vineyards, and 19 of them were found to be of the Kalmusia variispora species. More specifically, in lines 95-96 of the revised manuscript, we mention that we collected 19 pycnidial isolates with an olivaceous color that were K. variispora, based on rDNA multigene alignment based on ITS, LSU, and SSU. Further genetic loci (tef1-a, rpb2, and b-tub) were sequenced for 12 out of the 19 K. variispora isolates that provided the final identification. The 12 isolates were selected to represent different sampling areas (provinces of Paphos, Limassol, and Nicosia) and V. vinifera cultivars, as shown in the Supplementary Table S1.

Comment 3 Table 1: it would be good to adjust the column width or the font size in such a way that all symbols of accession numbers remained within one row. Table title: I suggest “Fungal species” would better suit to this table than “Fungal taxa”.

Response 3: We have replaced “Fungal taxa” with “Fungal species” in the title of Table 1, and we have made the appropriate rearrangements to fit all symbols of accession numbers within one row in the revised manuscript.

Comment 4 Line 201-202: please, explain the sentence about pine needles. Why did you use them to observe pycnidia formation? Is this a common method?

Response 4: This is a commonly used method to induce the formation of sexual and or asexual reproductive structures in ascomycetes. More specifically, sterilized plant parts in combination with techniques, such as exposure to cool-white, fluorescent light and or UV light radiation, were found effective in successfully inducing sporulation (Crous et al. 2006, Wulandari et al. 2009, Glienke et al. 2011). Furthermore, the formation of pycnidia or other structures on the pine needles helps us to pick these structures free of culture media or other contaminants to proceed with their microscopic analyses.

Comment 5 Line 413: I suggest there should be CBS 151331 instead of CBS 151221.

Response 5: This is a typo (CBS 151331 instead of CBS 151221) and we have corrected it accordingly in the revised manuscript (lines 232, 404).

Comment 6 Lines 508-510: CBS 151329, which showed the lack of laccase-degrading enzymes, also showed the lowest pathogenicity, while CBS151325, which showed the maximum pathogenicity, also demonstrated a significant laccase-degrading activity. Did you make a correlation analysis between these two traits? It would be good to illustrate these lines of the Discussion section with this data.

Response 6: The following text has been added in the revised manuscript (lines 491-498): “More specifically, the more pathogenic isolates CBS 151325, CBS 151327, and CBS 151331 (Figure 5) exhibited more pronounced laccase production (Figure 3), compared to CBS 151334, CBS 151330, and CBS 151329, which were less aggressive and faint or no laccase producers. Thus, laccase production by K. variispora could be qualitatively related to pathogenicity in terms of wood discoloration and lesion length on lignified grapevine canes. In the present study, the observed variability in ligninase production, where one isolate was not able to produce lignin-degrading enzymes, provides insights into the diverse enzymatic strategies of K. variispora”. Since laccase production was not quantitatively estimated, a correlation analysis with pathogenicity estimates (wood discoloration and lesion length in centimeters) was not possible.

Comment 7 Line 525-528: I do not find any author who did investigation itself. Please, check and correct.

Response 7: We agree with your suggestion. Unfortunately, we forgot to include the responsible co-authors for the investigation by mistake. In the revised manuscript, we included this important information (line 508).

Comment 8 References: please, check for the correspondence to the journal rules. For example, compare Ref. 1 and Ref 2 (placement of the year of publication).

Response 8: We agree with your comment, and we have formatted the references according to the journal’s rules in the revised manuscript.